In silico prediction of candidate gene targets for the management of African cassava whitefly (Bemisia tabaci, SSA1-SG1), a key vector of viruses causing cassava brown streak disease

http://orcid.org/0000-0002-0577-6460 Kaweesi Tadeo 1 2 3 karln1844@yahoo.com
Colvin John 1
http://orcid.org/0000-0002-6405-9162 Campbell Lahcen 4
Visendi Paul 5
Maslen Gareth 4
http://orcid.org/0000-0002-4489-3133 Alicai Titus 3
Seal Susan 1
1 Natural Resources Institute, University of greenwich , Chatham Maritime, Kent , United Kingdom
2 Rwebitaba Zonal Agricultural Research and Development Institute, National Agricultural Research Organization , Fort Portal, Kabarole , Uganda
3 National Crops Resources Research Institute, National Agricultural Research Organization , Kampala , Uganda
4 Wellcome Genome Campus, European Molecular Biology Laboratory, European Bioinformatics Institute (EMBL-EBI) , Hinxton, Cambridge , United Kingdom
5 Institute for Molecular Bioscience, Australian Research Council Centre of Excellence for Innovations in Peptide and Protein Science, The University of Queensland, Queensland University of Technology , Brisbane, Queensland , Australia
Shah Anis
Electronic publication date: 2024 Feb 23
Publication date: 2024
Volume: 12
Electronic Location ID: e16949
Received 2023 Oct 16; Accepted 2024 Jan 24
Copyright: © 2024 Kaweesi et al.
Copyright year: 2024
Copyright holder: Kaweesi et al.
License: This is an open access article distributed under the terms of the Creative Commons Attribution License, which permits unrestricted use, distribution, reproduction and adaptation in any medium and for any purpose provided that it is properly attributed. For attribution, the original author(s), title, publication source (PeerJ) and either DOI or URL of the article must be cited.
License URL: https://creativecommons.org/licenses/by/4.0/

Keywords: Bemisia tabaci, Gene targets, Osmoregulation, Sucrase, Aquaporin, Alpha-glucosidase, Symbiosis, Portiera aleyrodidarum, RNA intereference, Pest management

Funding: Natural Resources Institute, University of Greenwich from a grant provided by the Bill & Melinda Gates foundation Grant Agreement OPP1058938 This research was funded by the Natural Resources Institute, University of Greenwich from a grant provided by the Bill & Melinda Gates foundation (Grant Agreement OPP1058938). The funders had no role in study design, data collection and analysis, decision to publish, or preparation of the manuscript.

==============================
Whiteflies (Bemisia tabaci sensu lato) have a wide host range and are globally important agricultural pests. In Sub-Saharan Africa, they vector viruses that cause two ongoing disease epidemics: cassava brown streak disease and cassava mosaic virus disease. These two diseases threaten food security for more than 800 million people in Sub-Saharan Africa. Efforts are ongoing to identify target genes for the development of novel management options against the whitefly populations that vector these devastating viral diseases affecting cassava production in Sub-Saharan Africa. This study aimed to identify genes that mediate osmoregulation and symbiosis functions within cassava whitefly gut and bacteriocytes and evaluate their potential as key gene targets for novel whitefly control strategies. The gene expression profiles of dissected guts, bacteriocytes and whole bodies were compared by RNAseq analysis to identify genes with significantly enriched expression in the gut and bacteriocytes. Phylogenetic analyses identified three candidate osmoregulation gene targets: two α-glucosidases, SUC 1 and SUC 2 with predicted function in sugar transformations that reduce osmotic pressure in the gut; and a water-specific aquaporin (AQP1) mediating water cycling from the distal to the proximal end of the gut. Expression of the genes in the gut was enriched 23.67-, 26.54- and 22.30-fold, respectively. Genome-wide metabolic reconstruction coupled with constraint-based modeling revealed four genes (argH, lysA, BCAT & dapB) within the bacteriocytes as potential targets for the management of cassava whiteflies. These genes were selected based on their role and essentiality within the different essential amino acid biosynthesis pathways. A demonstration of candidate osmoregulation and symbiosis gene targets in other species of the Bemisia tabaci species complex that are orthologs of the empirically validated osmoregulation genes highlights the latter as promising gene targets for the control of cassava whitefly pests by in planta RNA interference.

Introduction

Bemisia tabaci sensu lato is a globally important whitefly pest affecting agricultural production by direct damage and as a vector of plant viruses (Jones, 2003). Collectively, members of the B. tabaci species complex transmit more than 300 plant viruses, some of which cause crop diseases that have been listed among the top 10 most economically damaging plant viruses, globally (Legg et al., 2006; Scholthof et al., 2011). To date, more than 39 morphologically indistinguishable species within the B. tabaci complex have been reported (Mugerwa et al., 2018). These species exhibit differences in fitness (Mugerwa et al., 2019) and adaptation to different agroecological areas. The focus of this study is a member of the B. tabaci species complex called Sub-Saharan Africa 1-subgroup 1 (SSA1-SG1).

SSA1-SG1 has a broad host plant range but, in recent years, populations have risen to “superabundant” levels (Legg et al., 2014a) on cassava crops in East Africa. Compounding these problems, SSA1-SG1 is a vector of viruses that cause cassava mosaic disease and cassava brown streak disease affecting the production of cassava (Burns et al., 2010). Cassava brown streak is caused by at least two genetically distinct virus species (Mbanzibwa et al., 2009; Ndunguru et al., 2015), and has been reported as one of the serious threats to the world’s food security (Pennisi, 2010). Total annual losses of cassava have been estimated at US$1.9–2.7 billion (Legg et al., 2006; Scholthof et al., 2011) in areas affected by B. tabaci-transmitted virus diseases. It is important, therefore, that efficient control methods are developed to manage the superabundant whitefly populations in different agro-ecologies.

To mitigate the spread of the disease, several options have been suggested. These include (i) implementing an efficient seed system (McQuaid et al., 2016), (ii) breeding for host resistance (Kawuki et al., 2016), (iii) phytosanitation (Legg et al., 2017), and (iv) development of efficient options for management of whitefly vector(s) (Legg et al., 2014b). Recently, several approaches have been considered in the management of whiteflies in cassava-producing areas. These include pesticide application (Omongo et al., 2022), cultural control and host resistance breeding (Parry et al., 2020), biological control (Tize et al., 2023), and application of botanical oils (Amour et al., 2023). However, conventional strategies against CBSD and the vector for its causative viruses have proven to be difficult to develop. There are also challenges in the implementation of management options against cassava whiteflies due to farmers’ perception of cassava as a low value crop (Legg et al., 2014b). Ongoing efforts are focused on the development of novel management options, such as transgenic approaches, targeting both the viruses and the whitefly vector. There has been progress in the development of RNAi approaches against cassava brown streak viruses (Patil et al., 2011). Based on these experiences, efforts have been geared towards development of precision methods against cassava whiteflies including in planta RNAi and CRISPR-Cas9 approach (Luo et al., 2017; Heu et al., 2020).

The success of transgenic approach depends on the selection of effective gene targets. Within the whitefly, both osmoregulation and amino acid provisioning are essential for the survival of the pest and hence provide potentially suitable gene targets. Whiteflies feed solely on plant phloem sap that contains high concentrations of soluble sugars, usually sucrose, at concentrations often exceeding 1M (Douglas, 2006a). As a direct consequence of its high sugar content, the osmotic pressure of phloem sap is considerably higher than the body fluids of the insect, and osmoregulatory failure in the insect is predicted to cause the net flow of body water to the gut lumen, resulting in dehydration and death. Whiteflies display specialized physiological mechanisms that enable them to utilize phloem-sap effectively. Since the osmotic pressure is determined by the molarity and not weight of solutes, phloem sap-feeders overcome the osmotic challenge by both water recycling and transforming much of the ingested sucrose into oligosaccharides of lower molarity (Rhodes, Croghan & Dixon, 1997). This makes the osmoregulation system, one of the most important systems to target in the management of whitefly in different agricultural systems. Osmoregulation has been studied in detail in the aphid Acyrthosiphon pisum Harris (Ashford, Smith & Douglas, 2000; Cristofoletti et al., 2003), demonstrating a key role for two gene products, α-glucosidase of the glycosyl hydrolase family 13 (EC 3.2.1.20) known as SUC1 and a water-carrying aquaporin, known as AQP1.

Whiteflies and other phloem-sap feeding insects also depend on endosymbionts for essential amino acid and B vitamin biosynthesis (Douglas, 2016; Liu et al., 2020). Whiteflies achieve this through an obligate interaction with the intracellular symbiont, Portiera aleyrodidarum (henceforth referred to as Portiera). Portiera is a vertically transmitted endosymbiont restricted to the cytoplasm of a specialized insect cell known as a bacteriocyte (Baumann, 2005; Luan et al., 2016). Its major role is to provide several essential amino-acids or metabolites for intermediate reactions within different essential amino acid biosynthesis pathways in the host. The exploitation of this interdependency in the control of B. tabaci populations requires a comprehensive knowledge of both B. tabaci and endosymbiont genetics and physiology. The increasing availability of genomic resources of different B. tabaci species (Campbell et al., 2023) and their respective endosymbionts offers opportunities to study this interaction at the systems level.

Genome-scale stoichiometric models coupled with constraints-based modelling techniques like flux balance analysis and flux variability analysis (Varma & Palsson, 1994) offer an overview of the gene-protein stoichiometric network and flow of flux, that can facilitate studying the interaction between B. tabaci and symbionts at a system level. Genome-scale metabolic models have been used to identify essential gene targets for pharmaceutical applications (Hu et al., 2007; Jamshidi & Palsson, 2007) and also for efficient production of amino acids in different organisms (Pharkya, Burgard & Maranas, 2003; Jin et al., 2007).

The goal of this study was to identify key gene targets for the management of cassava whitefly, by targeting the most critical systems in the B. tabaci member species SSA1-SG1. The identified genes and procedures should facilitate application of RNA interference in the management of this economically important species and closely related B. tabaci member species. This in silico analysis was assisted by the availability of draft genome sequences for three B. tabaci species, MEAM1 (Bemisia argentifolii) (GenBank accession number: PRJNA312470), SSA-ECA (GenBank accession number SRP125415) (Chen et al., 2016, 2019), accessed through a whitefly genome database (http://www.whiteflygenomics.org) and SSA1-SG1-Ug (Campbell et al., 2023). Our strategy was based on identifying genes with enriched expression in gut and bacteriocytes, key organs for osmoregulation and symbiosis respectively. This was coupled with phylogenetic analysis of other insects, including other B. tabaci species for identification of osmoregulation genes and application of genome-scale metabolic reconstruction and constraint-based modelling to identify key symbiosis genes.

Materials and Methods

Sample collection and RNA isolation

The cassava B. tabaci population (SSA1-SG1-Ug) was derived from a single virgin female and male collected from Namulonge, Uganda. Namulonge is one of the areas in Uganda with superabundant B. tabaci populations and a high incidence of both cassava brown streak disease and cassava mosaic disease (Mugerwa et al., 2018). The B. tabaci SSA1-SG1-Ug population (member species name ‘SSA1-SG1’) was maintained at the Natural Resources Institute (UK), on aubergine (Solanum melongena cv. Black Beauty) under quarantine insectary conditions (60% relative humidity, 28 °C and 12 h light:12 h dark regime) with light intensity of 500 µmol m−2 s−1 PAR (Photosynthetic Active Radiation). Approximately 10,000 bacteriocytes and 300 guts were dissected in phosphate-buffered saline (PBS, 1X) from 500 and 300 female adult whiteflies respectively using fine pins and a stereomicroscope (Fig. 1). Gut samples were stored in RNAlaterTM stabilization solution (Invitrogen, Waltham, MA, USA) to preserve their RNA quality. A total of 50 female adult whiteflies were also collected in three replicates, flash-frozen in liquid nitrogen and stored at −80 °C prior to analysis of whole body gene expression in these insects.

Figure 1 Bacteriocytes dissected from the abdomen of Bemisia tabaci (SSA1-SG1).

(A) The location of the bacteriocytes in the abdomen of the whitefly and (B) Dissected gut (translucent tube-like organ) and bacteriocyte (yellow cells) harboring the primary endosymbiont.

RNA extraction from dissected guts and RNAseq library preparation

Total RNA was isolated from B. tabaci gut, bacteriocytes and whole body samples using TRIzoLTM reagent (Invitrogen), with slight modifications made in the manufacturer’s instructions to ensure purity and quantity of RNA for downstream analysis for low tissue amounts. Specifically, we included a second clean-up with 100 µL chloroform, a second phase separation steps and addition of 1 µl linear acrylamide to the aqueous phase (before the isopropanol step) to improve the precipitation of RNA. Total RNA was further treated with ezDNaseTM enzyme (Invitrogen) according to the manufacturer’s instructions, to remove contaminating genomic DNA. RNA quantity and integrity were confirmed using a Qubit fluorometer (Life Technologies, Carlsbad, CA, SA).

Total RNA was depleted of cytoplasmic and mitochondrial rRNA using the Ribo-Zero Gold rRNA removal epidemiology kit (Illumina, San Diego, CA, USA) according to manufacturer’s instructions. For the whole body samples, 1 µg total RNA was used for rRNA depletion, while 367 and 230 ng total RNA were used for the bacteriocyte and gut samples. RNAseq libraries were prepared with the NEBNext Ultra II RNA Library Prep Kit (New England Biolabs Inc, Ipswich, MA, USA), according to manufacturer’s instructions. For the whole body samples, two technical replicates of 15 ng rRNA-depleted RNA were used as input while for the bacteriocyte and gut samples, all the rRNA-depleted RNA (not quantified) was used as input. The final libraries were quantified with a Qubit HS DNA assay (Thermo Fisher Scientific, Waltham, MA, USA) while the size distribution was determined using a Fragment Analyzer instrument (Agilent Technologies, Inc., Santa Clara, CA, USA). Equimolar amounts of each library were pooled for sequencing.

RNA sequencing and RNAseq analysis

The RNAseq libraries were sequenced and >40 million single-end 75 nt reads were generated on a NEXTseq500 sequencing platform at the Cornell Genomics Facility (Biotechnology Resource Center). Raw reads were filtered to remove low-quality reads and adaptor sequences with Cutadapt (Martin, 2011). The cleaned RNAseq reads were aligned against the SSA1-SG1-Ug genome (GCA_902825415.1) using HISAT2 (Kim, Langmead & Salzberg, 2015), which runs Bowtie2 in the background. The accepted hits were used, firstly to develop gene annotation files containing gene models based on RNAseq aligned reads for guts, bacteriocytes and whole body using cufflinks and cuffmerge version 2.2.1 (Trapnell et al., 2010). The accepted hits were then assembled and aligned reads counted using featureCounts (Liao, Smyth & Shi, 2014). Differential gene expression in cassava B. tabaci (SSA1-SG1-Ug) gut or bacteriocytes relative to the whole body was performed based on feature counts using edgeR (Robinson, McCarthy & Smyth, 2009) version 3.28.0 with three considerations. These include (i) false discovery rate controls based on the Benjamin and Hochberg method (Benjamini & Hochberg, 1995), (ii) trimmed mean of M-values normalization to account for library size variation between the samples and (iii) normalization for each gene, where the counts for each gene were mean centred and scaled to unit variance. The statistical power of this experimental design as calculated in RNAseqPower (Hart et al., 2013) was one, with a minimum sample size of eight whiteflies per sample to detect a two-fold difference in gene expression. Genes with enriched or depleted expression in the gut samples were identified by the general linear model (likelihood ratio test) analysis of log2 fold difference (logFC) data.

Identification of osmoregulation genes

Nucleotide sequences for the genes enriched in B. tabaci guts were extracted from the SSA1-SG1-Ug genome (GCA_902825415.1) using Samtools faidx tool (Li et al., 2009). Blastx search against NCBI protein reference non-redundant database (nr) with a cutoff E-value of 1e-3 (Altschul et al., 1997) mapping and annotation of these sequences were done using Blast2GO suite (Conesa et al., 2005). The suite was also used to generate and characterize gene ontology terms based on BLAST output. Gene ontology annotation was characterized into three groups, namely molecular function, biological process and cellular component. Selected sequences with metabolic functions were further translated to protein sequences using the ExPASy molecular biology server of the Swiss Institute of Bioinformatics (Gasteiger et al., 2003). The translated protein sequences with Aamy domain (PF00128, IPR006047) for glycosyl hydrolases (family 13) and aquaporin domain (PF00230, IPR000425), for osmoregulation genes were selected from the blast2go output. The selected sequences were further analyzed using SignalP 4.0 server (Petersen et al., 2011), NCBI conserved domain database (Marchler-Bauer et al., 2015), HMMER V.3 (Eddy, 2011) and InterProScan V5.52-86.0 (Zdobnov & Apweiler, 2001) to confirm the functional domain for sucrase and aquaporin. The selected protein sequences were also analyzed using NCBI Batch web CD-search tool to determine if the conserved domains were complete or incomplete at N-terminus or C-terminus.

Phylogenetic analysis

Phylogenetic analyses compared selected sequences to experimentally validated osmoregulation genes in other B. tabaci species and aphids (NCBI accession number: NP_001119607.1). Protein sequences were aligned by ClustalW (Larkin et al., 2007) implemented in MEGA X using default parameters. The best-fit amino acid substitution model to describe the substitution pattern was selected based on Bayesian information criterion (BIC) scores, computed using MEGA X (Kumar et al., 2018). The best amino acid substitution model with the lowest BIC score was selected and used in the phylogenetic reconstruction implemented in Bayesian Evolutionary Analysis Sampling Trees (BEAST version 1.10.2) (Suchard et al., 2018). BEAST was run with four gamma categories and strict molecular clock model at 1.0 clock rate. Yule model with uniform birth rate (Gernhard, 2008) and inverse gamma were used to provide prior on the tree. Markov Chain Monte Carlo (MCMC) was set with a chain length of 10,000,000. The generated tree was visualized using FigTree V1.4.3. The phylogenetic trees were used to study the relationship of selected genes with experimentally validated osmoregulation genes in other B. tabaci and aphid species (Price et al., 2007; Shakesby et al., 2009) to select the most suitable gene target.

Genome-scale reconstruction of Portiera and SSA1-SG1 metabolic models

Single compartment models for Portiera and B. tabaci SSA1-SG1 were reconstructed following the procedures of Thiele & Palsson (2010). A pseudo reaction for biomass generation was developed as the objective function for the model as described by Ankrah, Luan & Douglasa (2017). Stoichiometric coefficients in this reaction describe the relative abundance of essential building blocks, focusing on amino acid synthesis (amino acids, ATP and water), that enables Portiera and B. tabaci to grow and survive (Table S4). This reaction was selected as the objective in the calculation of optimal flux distribution for the Portiera and B. tabaci SSA1-SG1 model.

The two single-compartment models generated (for Portiera and B. tabaci) were combined to form a two-compartment model of B. tabaci SSA1-SG1 and Portiera, which was used for analysis and identification of essential symbiosis genes. The reactions in single models were renamed with a prefix SSA1_Por for Portiera model and SSA1_Bt for the B. tabaci SSA1-SG1 model. Metabolites in Portiera and B. tabaci models were localized in (p) and (c) compartments, respectively. Transport reactions were assigned to ensure that dead-end metabolites from each compartment are exchanged between the two compartments and import essential metabolites into the compartment. Exchange reactions were also included so that the metabolites or products produced by the two compartments are exported to the external environment (B. tabaci hemolymph).

Analysis and evaluation of models

Several parameters of the model were assessed. These included the optimal steady-state flux profile and essential reaction/genes in the Portiera and Portiera/B. tabaci model that would best predict the amino acid requirement for growth of Portiera and B. tabaci respectively. The model was analysed using flux balance analysis (Varma & Palsson, 1994) to determine the steady-state flux profile that optimizes the objective function and to simulate gene essentiality of the two-compartment model.

The amino acid requirement for growth of Portiera and B. tabaci (SSA1-SG1) were determined using flux balance analysis (FBA), with simulations performed under aerobic conditions with a maximum oxygen uptake rate of 20 mmol gDW−1 h−1. The objective function for the two-compartment model was developed (Table S4) to optimize the amino acid requirement for both Portiera and B. tabaci. FBA was also used to determine the effect of gene/reaction deletion on the metabolic model. Each reaction was perturbed, one at a time, by constraining both the upper and lower bound to zero, thereby restricting it from carrying flux. Gene or reaction knock out were implemented and growth simulated using SingleGeneDeletion or SingleReactionDelection function of COBRA Toolbox respectively. A reaction was classified as essential or critical if when removed from the model, the resultant value of biomass flux was zero (the network does not support growth) or non-essential if perturbation caused no effect on the network (network supports growth). Genes mediating reactions in essential amino acid biosynthesis were selected from the set of identified essential genes as potential gene targets for controlling cassava B. tabaci through RNAi. Minimization of metabolic adjustment (MOMA) (Segrè, Vitkup & Church, 2002) and regulatory on/off minimization (ROOM) (Shlomi, Berkman & Ruppin, 2005) were also used to confirm the essential genes selected.

Validation of selected genes using real-time quantitative PCR

The expression of the selected genes in the whitefly gut or bacteriocytes was validated using real-time quantitative PCR (qPCR). In brief, three technical samples of guts from 300 whiteflies, bacteriocytes from 500 whiteflies and whole body samples from 50 whiteflies were collected and preserved immediately at −80 °C. Total RNA was extracted as described above and its quantity and integrity were confirmed using a Qubit fluorometer (Life Technologies). Only samples with RIN values between 6–8 were considered for further analysis. A total RNA for whitefly gut, bacteriocyte, and whole body samples were normalized to 120 ng, to ensure an equal amount of starting RNA template. Normalized RNAs were then treated with 1 µl ezDNaseTM enzyme (Invitrogen) for 2 min at 37 °C to remove genomic DNA. The RNA sample was further incubated at 55 °C for 5 min with 10 mM DTT to inactivate the enzyme. First-strand cDNA synthesis was carried out using SuperScript™ III Reverse Transcriptase (Invitrogen) according to the manufacturer’s instructions. A three-step thermocycling protocol was performed on C1000 Thermocycler with CFX96 Real-time detection system (Bio-Rad, Hercules, CA, USA) with CFX Manager Version 3.1.1517.0823. The three steps included a polymerase activation and DNA denaturation at 95 °C for 3 min followed by 40 cycles of both denaturation at 95 °C and annealing/extension at 55.2 °C for 45 s. Both 60S ribosomal protein L13a (RPL13) and β-tubulin were used as the internal controls (Fig. S5, Table S5). No template controls (NTC) were also added to evaluate the quality of analysis. A total of 20 µl reaction mix containing 10 µl iQ™ SYBR® Green Supermix (2x), 1 µl forward and reverse primer (Table S1), 1 µl cDNA and 7 µl water was used. Melting curve analysis at the end of the reaction was done by increasing temperature from 55–95 °C in increments of 0.5 °C every 5 s to assess the dissociation characteristics of double-stranded DNA during heating (Table S5).

Results

Expression patterns of the gut and bacteriocytes of Bemisia tabaci SSA1-SG1

RNAseq analysis was conducted to identify candidate osmoregulation and symbiosis genes with enriched expression in the whitefly gut and bacteriocytes respectively. A total of 49,426,054, 47,305,213 and 47,646,265 raw reads was generated for the whitefly gut, bacteriocyte and whole body samples respectively. These samples had closely similar percentage mapping against the B. tabaci SSA1-SG1-Ug genome of 78.1, 83.84 and 78.0% for the gut, bacteriocyte and whole body samples, respectively. RNAseq analysis revealed a total of 10,027 transcripts (64% of what has been reported in B. tabaci species Middle East Asia Minor 1 (MEAM1)) (Chen et al., 2016). Of these transcripts, 1,316 and 1,909 had enriched expression in the whitefly gut and bacteriocytes relative to the whole body. This gene set was then assessed for candidate genes with osmoregulatory function, specifically genes encoding sucrase enzymes (SUC genes) and water-specific aquaporin channels (AQP genes) and symbiosis genes (amino acid synthesis, transport and horizontally transferred genes).

Genes encoding sugar metabolism with enriched expression in the SSA1-SG1 gut

A total of 24 transcripts encoding sugar processing enzymes with an AmyAc_Maltase (cd11328) catalytic domain and belonging to glycosyl hydrolase family 13 (EC 3.2.1.20) had significantly enriched expression in whitefly gut relative to the whole body (Table 1). Of these, seven transcripts had the highest expression with log2CPM ranging from 10.18–11.86, equivalent to a 17–30-fold enrichment in the B. tabaci SSA1-SG1 gut relative to whole body. Further analysis revealed that 14 of the 24 α-glucosidase genes with enriched expression in the gut had both a catalytic site residue, required for enzymatic activity, and a predicted signal peptide, required for the enzyme to function as an extracellular enzyme in the gut lumen (Table 1). Phylogenetic analysis of the 14 selected protein sequences identified three clusters (Fig. 2), with some protein sequences aligning most closely with the experimentally validated pea aphid sucrase protein (SUC1). The protein of SSA1-SG1-Ug that aligned most closely with the pea aphid SUC1 is ENSSSA1UGG000718 (SSA-ECA-Ssa05164), which has a predicted signal peptide and intact active site. The transcript was enriched 26-fold in the gut relative to the whole body. We predict that, as for the pea aphid SUC1, this protein has a key osmoregulatory function. We name the protein SSA1_SUC1 and its cognate gene SSA1_SUC1. Another protein that aligned closely to the experimentally validated pea aphid sucrase protein and its corresponding gene and had a high gene expression in the SSA1-SG1 gut was ENSSSA1UGG009974 (Ssa05154). We name this protein SSA1_SUC2 and its cognate gene SSA1_SUC2.

Table 1 Genes encoding proteins with characteristic catalytic site residue and a signal peptide for α-glucosidase enzyme.

Gene ID	Gene†	LogCPM	Fold
difference	Signal peptide	D-Score‡	Catalytic site residue§	
Cleavage
site	Nucleophile	Proton donor	
ENSSSA1UGG005530	Bta06059	1.05	52.71	+	0.760	A(19)F	D(224)	E(289)	
ENSSSA1UGG000414	Bta07377	8.14	30.06	+	0.768	S(20)R	D(227)	E(295)	
ENSSSA1UGG008427	Bta14419	10.44	29.65	+	0.474	S(20)H	D(230)	E(297)	
ENSSSA1UGG005077	Bta05340	2.04	28.25	+	0.739	G(21)H	D(183)	E(250)	
ENSSSA1UGG009590	Bta09633	4.35	28.25	+	0.774	C(20)N	D(227)	E(295)	
ENSSSA1UGG013152	Bta04298	10.09	27.67	–	0.195	–	D(19)	E(86)	
ENSSSA1UGG000718	Bta03818	9.43	25.99	+	0.843	C(16)R	D(228)	E(296)	
ENSSSA1UGG004292	Bta07764	7.35	25.46	+	0.512	E(20)E	D(257)	E(289)	
ENSSSA1UGG002701	Bta12682	11.23	25.11	–	0.119	–	D(288)	E(344)	
ENSSSA1UGG002293	Bta03439	10.18	24.93	+	0.833	A(22)Q	D(227)	E(232)	
ENSSSA1UGG012024	Bta08425	9.96	23.92	+	0.825	G(20)I	D(231)	E(290)	
ENSSSA1UGG009983	Bta05396	9.93	23.43	–	0.329	–	D(244)	E(312)	
ENSSSA1UGG005217	Bta08427	7.12	22.63	+	0.860	G(20)V	D(227)	E(286)	
ENSSSA1UGG009974	Bta05386	11.05	21.71	+	0.743	A(24)V	D(268)	E(337)	
ENSSSA1UGG007699	Bta06458	9.12	20.25	+	0.765	G(20)G	E(229)	R(287)	
ENSSSA1UGG001314	Bta09696	5.07	20.25	+	0.742	A(25)Q	–	–	
ENSSSA1UGG005058	Bta14422	9.04	17.27	+	0.799	Q(23)S	D(228)	E(296)	
ENSSSA1UGG013610	Bta10022	4.74	17.27	–	0.147	–	–	–	
ENSSSA1UGG000756	Bta07452	8.99	11.00	–	0.428	–	D(227)	E(294)	
ENSSSA1UGG003676	Bta11358	2.71	8.51	+	0.810	A(20)D	–	–	
ENSSSA1UGG005127	Bta15649	11.25	17.15	+	0.693	E(21)F	D(233)	E(300)	
ENSSSA1UGG000967	Bta01478	7.17	7.36	–	0.105	–	D(281)	E(369)	
ENSSSA1UGG003278	Bta13914	3.35	4.41	+	0.805	S(17)T	D(220)	E(288)	
NP_001119607.1_Aphid	–	–	–	+	0.781	S(21)E	D(236)	E(304)	
Notes:

† Ortholog in Bemisia tabaci MEAM1.

‡ Discrimination score (D-score) for a best-predicted signal peptide for predicted proteins of GH13 family.

§ Catalytic site residue in each GH13 sequence. The position of the residue is indicated by the number in the parentheses. Presence or lack of signal peptide denoted as + or – respectively.

Figure 2 Alpha-glucosidases in B. tabaci SSA1-SG1.

Phylogenetic relationships of α-glucosidases in B. tabaci SSA1-SG1 that have predicted signal peptides and catalytic site residue (Ensembl “stable ID”: ‘ENSSSA1SG1TXXXXX’) and the experimentally validated sucrase gene in Acyrthosiphon pisum (NCBI accession number: NP001119607.1). Phylogenetic analysis was done using Bayesian method implemented in Bayesian Evolutionary Analysis Sampling Trees (BEAST version 1.10.2).

Water cycling within cassava Bemisia tabaci SSA1-SG1 gut

Seven aquaporin genes were identified in the transcriptome of B. tabaci SSA1-SG1-Ug (Table 2). The transcript abundance of three genes (ENSSSA1UGG010352 (SSA-ECA-Ssa03484), ENSSSA1UGG011821 (SSA-ECA-Ssa02238) and ENSSSA1UGG001892 (SSA-ECA-Ssa08000) were significantly enriched in the gut with log2CPM of 8.49, 5.17 and 3.31 and fold difference of 32.7, 2.2 and 2.1-fold respectively. Phylogenetic analysis (Fig. 3) included the sequences of experimentally-validated AQP proteins in B. tabaci studied by Van Ekert et al. (2016), revealing that ENSSSA1UGG010352 has homology to the water channel BtDRIP. Inspection of the predicted protein sequence of ENSSSA1UGG010352 confirmed that it had the motifs of a DRIP water channel: the two NPA boxes, the ar/R constriction region with phenylalanine at position 71, histidine at position 198, alanine at position 207 and arginine at position 213, and a mercury-sensitive cysteine residue located two residues after the first NPA box.

Table 2 Aquaporin sequences variation in transcriptome of Bemisia tabaci SSA1-SG1.

Gene ID	NCBI-Blast	Gene‡	Ar/R Filter	NPA signature motif	
Loop B	Loop E	
ENSSSA1UGG010352	Aquaporin 1	Bta01973†	F (71) H (89) A (207) R (213)	NPA (91)	NPA (210)	
ENSSSA1UGG000288	Aquaporin 2A	Bta13786	F (183) H (201) A (319) R (325)	NPA (204)	NPA (322)	
ENSSSA1UGG011828	Aquaporin 5	Bta07507	F (63) I (80) S (199) R (205)	NPA (82)	NPA (202)	
ENSSSA1UGG011821	Aquaporin 6	Bta07504	F (139) H (157) A (280) R (286)	NPA (159)	NPA (283)	
ENSSSA1UGG005288	Aquaporin AQPAe.a	Bta03161	F (61) H (80) T (195) R (201)	NPA (82)	NPA (198)	
ENSSSA1UGG001892	Aquaporin 7 (AQP 12L)	Bta14320	–	–	–	
Notes:

† AQPcic-like aquaporin (NCBI accession no. EU127479.1), experimentally validated water-specific gut aquaporin gene of whitefly (Mathew et al., 2011).

‡ Corresponding gene in B. tabaci MEAM1.

Figure 3 Aquaporin genes of B. tabaci SSA1-SG1.

Phylogenetic relationship of aquaporin genes of B. tabaci SSA1-SG1 (ENS numbers) and experimentally validated aquaporins in B. tabaci MEAM1 studied in ref 20 (APA numbers). Phylogenetic analysis used the Bayesian method implemented in Bayesian Evolutionary Analysis Sampling Trees (BEAST version 1.10.2).

Genes with enriched expression in the cassava Bemisia tabaci bacteriocytes

Interaction between B. tabaci and Portiera for essential amino acid provisioning takes place within the bacteriocyte. RNASeq analysis to identify genes with enriched expression in SSA1-SG1 bacteriocytes relative to whole body revealed a total of 1,909 transcripts with enriched expression log fold change (log FC) >1 in the bacteriocyte. Of these, 20 transcripts/genes encoding for proteins involved in amino acid biosynthesis and transport were selected (Table 3). Another category of genes with enriched expression in the bacteriocyte was the horizontally transferred genes (HTGs). HTGs contributing to lysine, arginine and proline biosynthesis within bacteriocytes had relatively high gene expression compared to other HGTs. Proteins encoded by these genes included; 2OG-Fe, oxygenase, allophanate hydrolase, 4-hydroxy-tetrahydrodipicolinate reductase (dapB), diaminopimelate decarboxylase (LysA), arginosuccinate lyase (ArgH) and arginosuccinate synthase (ArgG) (Table 4). The role and essentiality of these genes in the genome-scale metabolic network was analysed using constraint-based metabolic modelling techniques.

Table 3 Genes for amino acid synthesis and transport with enriched expression in the cassava whitefly (SSA1-SG1).

Gene ID	NCBI-BLAST Hit	Domain	LogCPM	Fold differences	MEAM1		
		Name	Interpro			P-value	Gene	
ENSSSA1UGG010036	Branched-chain-amino-acid aminotransferase	Aminotrans_4	IPR001544	8.89	6.79	0	Bta10673	
ENSSSA1UGG011355	Kynurenine--oxoglutarate transaminase 3	KAT_III	IPR034612	8.41	4.75	0	Bta05157	
ENSSSA1UGG005916	Phosphoserine aminotransferase	Aminotran_5	IPR000192	7.30	6.16	3.7e-85	Bta04363	
ENSSSA1UGG006055	y+L amino acid transporter 2	AA_Permease_2	IPR002293	7.39	8.59	0	Bta11657	
ENSSSA1UGT030948	High affinity cationic AA transporter 1	AA_Permease_2	IPR002293	6.57	3.19	1.01	Bta01499	
ENSSSA1UGG001116	Cationic amino acid transporter 3	AA_Permease_C	IPR029485	6.36	4.07	1.0e-105	Bta11613	
ENSSSA1UGG009631	y+L amino acid transporter 2	AA_Permease_2	IPR002293	5.98	3.21	3.5e-258	Bta02775	
ENSSSA1UGG009936	y+L amino acid transporter 2	AA_Permease_2	IPR002293	5.96	5.24	0	Bta08771	
ENSSSA1UGG010202	High affinity cationic AA transporter 1	AA_Permease_2	IPR002293	5.18	12.25	0	Bta13409	
ENSSSA1UGG005615	Transmembrane protein 104 homolog	Aa_trans	IPR013057	7.41	5.32	0	Bta03456	
ENSSSA1UGG000281	Proton-coupled amino acid transporter 1	Aa_trans	IPR013057	4.83	4.49	1.4e-202	Bta08775	
ENSSSA1UGG000876	Proton-coupled amino acid transporter 4	Aa_trans	IPR013057	4.74	3.25	5.4e-127	Bta08693	
ENSSSA1UGG011915	Proton-coupled amino acid transporter 1	Aa_trans	IPR013057	4.88	4.78	1.3e-222	Bta01726	
ENSSSA1UGG000481	Sodium-coupled neutral AA transporter 9	Aa_trans	IPR013057	2.66	2.47	7.8e-21	Bta11166	
ENSSSA1UGG011933	Proton-coupled amino acid transporter 1	Aa_trans	IPR013057	0.61	3.62	9.6e-08	Bta01718	
ENSSSA1UGG008679	Vesicular glutamate transporter 3	Major facilitator	IPR020846	6.95	11.97	0	Bta06520	
ENSSSA1UGG004279	Neutral and basic amino acid transport protein rBAT	GH 13	IPR006589	6.26	2.92	2.2e-246	Bta10614	
ENSSSA1UGG010267	excitatory amino acid transporter 1	PTR2	IPR000109	5.02	7.09	4.3e-317	Bta11487	
ENSSSA1UGG011338	peptide transporter family 1	PTR2	IPR000109	3.40	8.99	0	Bta04879	
ENSSSA1UGG006371	Na-dependent excitatory AA transporter glt-6	SDF	IPR001991	2.01	4.02	5.4e-28	Bta09439	

Table 4 Horizontally transferred genes of bacterial origin with enriched expressed in cassava B. tabaci (SSA1-SG1) bacteriocytes.

Gene ID	Description	Pathway	Gene expression	Possible origin: Bacteria		
			LogCPM	Fold difference	Genus	Gene***	
ENSSSA1UGG005107	2OG-Fe (II) oxygenase	Arg, Proline	11.16	1.00	Pantoea	Bta20012	
ENSSSA1UGG004185	4-hydroxy-tetrahydrodipicolinate reductase, dapB	L-lysine	7.88	5.45	Rickettsia	Bta20020	
ENSSSA1UGG003633	Diaminopimelate decarboxylase, LysA	L-lysine	6.89	13.89	Planctomyces	Bta03593	
ENSSSA1UGG006878	Diaminopimelate epimerase, DapF	L-lysine	0.13	1.57	Klebsiella	Bta06657	
ENSSSA1UGG006186	ATP-dependent dethiobiotin synthetase, BioD	Biotin	6.69	7.11	Wolbachia	Bta01938	
ENSSSA1UGG001672	Arginosuccinate lyase, ArgH	L-arginine	6.68	1.79	Erwinia	Bta00063	
ENSSSA1UGG001722	Arginosuccinate synthase, ArgG	L-arginine	5.95	2.10	Pantoea/Erwinia	Bta00062	
ENSSSA1UGT031701	Biotin synthase, BioB	Biotin	5.21	1.41	Not well defined	Bta09725	
ENSSSA1UGG006186	ATP-dependent dethiobiotin synthetase, BioD	Biotin	5.16	−1.50	Wolbachia	Bta00840	
ENSSSA1UGG012028	Chorismate mutase, CM	shikimate	5.01	2.57	Rahnella	Bta15103	
ENSSSA1UGG001160	3-methyl-2-oxobutanoate hydroxymethyltransferase	Pantothenate	3.57	30.50	Pseudomonas	Bta05339	
Note:

*** Ortholog in MEAM1.

Metabolic models for Portiera and its host, SSA1-SG1

Single compartment models of both the whitefly and the primary endosymbiont were reconstructed. The reconstructed single compartment metabolic model for Portiera (SSA1-SG1) comprised of 76 intercellular reactions, 150 metabolites and a total of 90 genes hence supporting a metabolic network, iKT90 while that of B. tabaci SSA1-SG1 (iKT330) comprised 251 metabolites and 233 intracellular reactions, mediated by 330 metabolic genes. The shared metabolic interaction between Portiera SSA1-SG1 and B. tabaci SSA1-SG1 involved a total of 420 metabolic genes mediating 310 intercellular reactions and an optimal flux balance solution of 0.39 (h−1), hence a two-compartment model iKT420.

Shared metabolic interaction between Portiera and Bemisia tabaci SSA1-SG1

Genome-wide reconstruction of a metabolic pathway for Portiera from SSA1-SG1 revealed that Portiera had genes that mediated the terminal reaction of three essential amino acids. These were (i) threonine, (ii) methionine and (iii) tryptophan, with reactions producing 0.101, 0.046 and 0.012 mmol gDW−1 h−1 flux, respectively. Furthermore, Portiera also contributes to the synthesis of metabolites for intermediate reactions of other essential amino acids. The bulk of reactions of the biosynthesis pathways of seven essential amino acids (arginine, histidine, lysine, phenylalanine, isoleucine, valine and leucine) are mediated by genes in Portiera. Metabolites produced in these reactions are exported across a symbiosome membrane into the bacteriocyte for amino acid biosynthesis. Metabolites (intermediate precursors) exported across the symbiosome membrane include, (i) 0.371 mmol gDW−1 h−1 N(Omega)-(L-Arginino) succinate for arginine biosynthesis, (ii) 0.049 mmol gDW−1 h−1 L-Histidinol phosphate for histidine biosynthesis, (iii) 0.1599 mmol gDW−1 h−1 LL-2,6-Diaminoheptanedioate for lysine biosynthesis, (iv) 0.2097 mmol gDW−1 h−1 phenylpyruvate for phenylalanine biosynthesis, (v) 0.1133 mmol gDW−1 h−1 (5)-3-methyl-2-oxopentanoate for isoleucine biosynthesis, (vi) 0.3107 mmol gDW−1 h−1 3-methyl-2-oxobutanoate for valine biosynthesis and (vii) 0.1653 mmol gDW−1 h−1 3-carboxy-4-methyl-2-oxopentanoate for leucine biosynthesis.

In addition, the single-compartment model of Portiera revealed that Portiera lacks genes that encode for metabolites required in the biosynthesis of non-essential amino acids (alanine, asparagine, aspartic acid, cysteine, glutamic acid, glutamine, glycine, proline, serine and tyrosine). It also requires eight host-derived metabolites for the biosynthesis of essential amino acids. These include, (i) 2, 3, 4, 5-tetrahydrodipicolinate, (ii) L-aspartate, (iii) alpha-D-ribose 5-phosphate, (iv) 5-methyltetrahydrofolate, (v) L-glutamine, and (vi) L-serine. In addition, Bemisia tabaci SSA1-SG1 imports D-fructose, homocysteine, thiamine diphosphate, riboflavin, pyridoxine 5-phosphate and nicotinate D-ribonucleotide from the insect hemolymph to the bacteriocyte. On the other hand, it exports 39 and 54 metabolites to hemolymph and Portiera respectively.

Terminal reactions in the essential amino acid biosynthesis pathway are important because they influence the provisioning of amino acid. The generated metabolic reconstructions revealed that the terminal reaction of isoleucine, leucine, valine, histidine and phenylalanine are all mediated by genes of intrinsic origin in the host bacteriocyte of SSA1-SG1. For example, for branched-chain amino acids, one enzyme, branched-chain amino acid transaminases (EC 2.6.1.42) encoded by BCAT, mediate the conversion of 3-methyl-2-oxobutanoate to valine, 4-methyl-2-oxopentanoate to leucine and (S)-3-Methyl-2-oxopentanoate to isoleucine in SSA1-SG1 whitefly.

Essential gene targets in different essential amino acid biosynthesis pathways in SSA1-SG1

A total of 270 reactions were characterized as essential based on the growth rate. Of these, 129 reactions were mediated by genes in Portiera while 141 reactions were mediated by genes in B. tabaci SSA1-SG1 (Fig. S2). Single gene deletion also revealed that seven of the ten biosynthesis pathways had essential genes mediating their terminal reactions. Therefore, these genes were characterized as essential genes in the provisioning of essential amino acids in the cassava B. tabaci SSA1-SG1. These include, (i) argH (EC 4.3.2.1) for arginine biosynthesis, (ii) lysA (EC 4.1.1.20) for lysine biosynthesis (Fig. 4), (iii) branched-chain amino acid transaminase (EC 2.6.1.42) for isoleucine, valine and leucine biosynthesis, (iv) hisD (EC 1.1.1.23) for histidine and (v) aspC (EC 2.6.1.58) for phenylalanine biosynthesis (Fig. 5). Of these, two genes (argH and lysA) are horizontally transferred genes of bacterial origin while three genes (BCAT, hisD and aspC) are of intrinsic origin (Fig. 6). All the selected reactions except PHETA1 for phenylalanine biosynthesis were mediated by the protein encoded by a single gene. Therefore, three factors were considered in the selection of the best candidate symbiosis gene targets for the management of cassava whitefly SSA1-SG1. These include, (i) indispensability of the gene, (ii) reaction mediated by a single gene and (iii) gene mediating a terminal reaction.

Figure 4 Lysine and arginine biosynthetic pathway in SSA1-SG1.

Biosynthetic pathway for two essential amino acids (i) lysine comprising of asp-L (Aspartate), 4pasp (4-Phospho-L-aspartate), aspsa (L-Aspartate 4-semialdehyde), 23dhdp (2,3-Dihydrodipicolinate), thdp (2,3,4,5-Tetrahydrodipicolinate), s12a6o (N-Succinyl-2-L-amino-6-oxoheptanedioate), s126da (N-Succinyl-LL-2,6-diaminoheptanedioate), 26dap-LL (LL-2,6-Diaminoheptanedioate), 26dap-M (Meso-2,6-Diaminoheptanedioate) and (ii) Arginine with the following metabolites; gln-L (L-Glutamine), cbp (Carbamoyl phosphate), orn-L (L Ornithine), Citr-L (L-Citrulline) and argSuc (N(omega)-(L-Arginino)succinate). Reactions with asterisks (**) represent reaction mediated by horizontally transfered genes.

Figure 5 Phenylalanine and tryptophan biosynthesis pathway for two essential amino acids.

Biosynthesis pathway for two essential amino acids. (i) Phenylalanine pathway involving chor (Chorismate), pphn (Prephenate), phpyr (Phenylpyruvate) and (ii) Tryptophan pathway consisting of the following metabolites: 3ig3p (C′-(3-Indolyl)-glycerol 3-phosphate), anth (Anthranilate), pran (N-(5-Phospho-D-ribosyl) anthranilate), indole and 2cpr5p (1-(2-Carboxyphenylamino)-1-deoxy-D-ribulose 5-phosphate). Reactions with asterisks (**) represent reactions mediated by horizontally transfered gene. Genes highlighted in red colour are the essential genes while those highlighted in black are non-essential genes.

Figure 6 Pseudogenization and gene it should be compensation in SSA1-SG1.

Conservation of different genes mediating critical reaction in different essential amino acid biosynthesis pathway in both B. tabaci SSA1-SG1 and MEAM1. Genes represented in red are horizontally transferred genes while those in blue and light blue are genes of intrinsic origin and pseudogenes respectively.

Validation of expression of selected genes in Bemisia tabaci guts and bacteriocytes using real-time quantitative PCR

In addition to the selected osmoregulation genes (AQP1, SUC1 and SUC2), genes mediating terminal reactions in the different amino acid pathways were selected for validation of gene expression in the bacteriocyte. A total of five genes were selected as critical gene targets with enriched expression in the bacteriocyte compared to the whole body. These were (i) ENSSSA1UGG003633: diaminopimelate decarboxylase (LysA), essential in lysine biosynthesis, (ii) ENSSSA1UGG001672: arginosuccinate lyase (ArgH), essential in arginine biosynthesis, (iii) ENSSSA1UGG010036: branched-chain-amino-acid aminotransferase (BCAT), critical in isoleucine, leucine and valine biosynthesis, (vi) ENSSSA1UGG004484: aspartate aminotransferase (AAT), essential in phenylalanine biosynthesis and (viii) ENSSSA1UGG004185: 4-hydroxy-tetrahydrodipicolinate reductase (dapB), essential in lysine biosynthesis. Results from RNAseq analysis and RT-qPCR were similar with fold differences in the gut ranging from 22.30 to 26.54-folds for RT-qPCR compared to 21.71 to 32.67-folds from RNAseq analysis (Fig. 7). In the bacteriocyte, gene expression for the selected gene ranged from 0.57 to 10.07-folds for RT-qPCR compared to 1.79 to 13.89-folds for RNAseq analysis.

Figure 7 Gene expression of selected gene targets in whitefly gut or bacteriocyte compared to whole body.

Quantitative real-time PCR gene expression of selected genes in the cassava whitefly (SSA1-SG1) gut and bacteriocytes relative to whole body. Gene ENSSSA1UGG010352 (AQP1), ENSSSA1UGG000718 (SUC1) and ENSSSA1UGG009972 (SUC2) were selected as critical osmoregulation genes with enriched expression in the whitefly gut while ENSSSA1UGG004484 (AAT), ENSSSA1UGG001672 (ArgH), ENSSSA1UGG010036 (BCAT), ENSSSA1UGG004185 (dapB) and ENSSSA1UGG003633 (LysA) were selected as essential symbiosis genes that mediate terminal reaction in the phenylalanine, arginine, leucine, isoleucine, valine and lysine biosynthesis pathways. The relative gene expression of the selected genes was normalized relative to the relative expression of two reference genes (ribosomal protein L13a and β-tubulin) using normalized expression ∆∆ Cq method (Livak & Schmittgen, 2001).

Discussion

Results from this study have revealed a total of seven essential gene targets (three osmoregulation and four symbiosis genes). Of the three, two candidate genes belonging to sugar transforming glycosyl hydrolase family 13 were selected as gene targets for disrupting sucrose hydrolysis in SSA1-SG1. One of these genes was, ENSSSA1UGG000718 (SSA-ECA-Ssa05164), encoding for a protein with high homology (97.5%) to NCBI accession AQU43105.1 (KX390871.1). AQU43105.1 is an α-glucosidase sequence selected by Jing et al. (2016) and used by Luo et al. (2017) in the study of osmoregulation in MEAM1. This sequence corresponded to the MEAM1 gene Bta03818, and it contains both signal peptide and catalytic site residue and was highly homologous to the experimentally validated sucrase (Price et al., 2007) (Fig. 2). The second gene target identified in this study was ENSSSA1UGG009974 (SSA-ECA-Ssa05154), an ortholog of Bta05386, an α-glucosidase in MEAM1. This gene (i) had high expression in the gut, (ii) possessed both signal peptide and a catalytic site residue and (iii) aligned closely to the experimentally validated SUC1 (Fig. 2) (Price et al., 2007). Further supporting the application of these genes as targets in an RNAi system to manage B. tabaci populations, both transcriptomic and RT-qPCR gene expression analysis of the selected genes (ENSSSA1UGG009974 and ENSSSA1UGG000718) showed that the selected genes had enriched expression in the B. tabaci gut.

Previously, putative α-glucosidase (NCBI accession KF377803.1, gene Bta07452) was proposed as a good RNAi target for the management of whiteflies (Raza et al., 2016). However, our study showed that Bta07452 lacks both signal peptide and catalytic site residue (Table 1), indicating that this gene is enzymatically non-functional, and may be a pseudogene. The presence of a signal peptide is important, because they are required for the enzyme to function as an extracellular enzyme in the gut lumen (Von Heljne, 1998). Compounding this concern, the sequence (KF377803.1) used to synthesize dsRNA against SUC1 in the forementioned research yielded no predicted siRNAs in the entire coding sequence of MEAM1 whitefly genome, as tested using siRNA-Finder (Lück et al., 2019).

Another osmoregulation gene ENSSSA1UGG010352 (Ssa03484) (AQP1) has the features predicted as mediator of water recycling in the gut of B. tabaci SSA1-SG1. Analysis of the protein sequence showed that this gene encodes a protein with a conserved NPA box (asparagine-proline-alanine motif) that ensures water selectivity and an aromatic/arginine region as a proton filter. These features give the AQP1 a two-stage filter that ensures water selectivity (French et al., 2001). Water selectivity in water-specific aquaporins is achieved by a specialized structure of the ar/R residue that comprises four amino acids: (i) Phe at helix 2, (ii) His at helix 5, (iii) Cys at LE1 and (iv) Arg at LE2, forming a narrow constriction region of 2.8 angstrom (1 angstrom ≈ 0.1nm) (Sui et al., 2001; Thomas et al., 2002). The amino acid at LE1 in ar/R filter of water-specific aquaporin varies in different insect species. For example in B. tabaci MEAM1, BtDRIP/BtAQP1, the conserved water selective ar/R selectivity filter, comprised of Phe 71, His 198, Ala 207 and Arg 213 (Van Ekert et al., 2016). Pairwise comparison with the corresponding ortholog in MEAM1 showed that this gene shared a 98.6% nucleotide similarity to Bta01973, an AQP1 in MEAM1. Downregulation of expression of this gene in the B. tabaci gut is predicted to disrupt the water recycling inducing osmotic collapse in the whiteflies.

For amino acid provisioning, genome-scale metabolic reconstruction coupled with constraint-based metabolic modeling showed that amino acid transaminase and horizontally transferred genes play a critical role in essential amino acid biosynthesis. Results presented here further showed that these horizontally transferred genes are placed at critical points within the essential amino acid biosynthesis pathways. For example, genes such as ENSSSA1UGG001672 (argH) and ENSSSA1UGG003633 (lysA) mediate terminal reactions of arginine and lysine respectively (Fig. 4). Constraint-based metabolic modeling showed that many of these innovative transfers within the whitefly encode for proteins that are essential for the development and survival of whiteflies. These innovative transfers (Husnik & McCutcheon, 2018) have allowed B. tabaci to slowly gain the genetic ability or new functionalities especially in the biosynthesis of essential amino acids and B vitamins. Furthermore, the acquisition of essential genes by the host within the bacteriocyte especially those that mediate terminal reactions enable the host not only to alter metabolite flux, but also to control both the synthesis and allocation of essential amino acids between the host and the endosymbiont. In this study, two pathways; L-lysine (Fig. 4) and L-phenylalanine (Fig. 4) demonstrate host compensation in the bacteriocyte. First, B. tabaci SSA1-SG1 acquired a bacterial gene dapB (EC 1.17.1.8) that encodes for 4-hydroxy-tetrahydrodipicolinate reductase in its lysine biosynthesis pathway. In SSA1-SG1-Portiera, dapB is missing (Figs. 4, 6) but the gene is still present in the Portiera genome of MEAM1-Portiera, though as a pseudogene (Luan et al., 2015). These results imply, (i) SSA1-SG1 is co-evolving with Portiera faster than MEAM1, or (ii) It also confirms the loss of essential genes especially by the endosymbiont (Rao et al., 2015) due to relaxed functional constraint arising from gene redundancy in the shared metabolic interaction within the bacteriocyte. Therefore, the loss of dapB in Portiera has enabled the horizontally acquired dapB gene in the bacteriocyte to take over the role of synthesizing 2,3,4,5-Tetrahydrodipicolinate from 2,3-Dihydrodipicolinate rendering the transferred gene indispensable in the biosynthesis of lysine in SSA1-SG1 bacteriocyte. Secondly, a similar compensation process is still on-going within the phenylalanine biosynthesis pathway. B. tabaci SSA1-SG1 acquired CM (EC 5.4.99.5) encoding chorismate mutase, an enzyme that converts chorismate to prephenate, a precursor of phenylpyruvate (Fig. 5). Essentiality analysis revealed that CM is dispensable. This is because two pathways can lead to prephenate synthesis, one mediated within Portiera and the other in the bacteriocyte. Speculatively, due to this redundancy, it can be predicted that pheA (EC 5.4.99.5), a gene that mediates prephenate biosynthesis within Portiera will undergo pseudogenization, making CM in the bacteriocyte an essential gene in the biosynthesis of phenylalanine.

Among the genes of intrinsic origin, branched-chain amino acid transferase (BCAT) was identified as an indispensable gene encoding for a protein that mediated the terminal reaction of branched-chain amino acids (BCAA) (isoleucine, leucine and valine) within the cassava B. tabaci (SSA1-SG1) bacteriocyte. The mediation of the terminal reaction of the three essential amino acids and its indispensability makes BCAT one of the best symbiosis gene targets. Therefore, it can be hypothesized that knockdown of BCAT will have a significant effect on the survival of B. tabaci SSA1-SG1. Several studies have demonstrated by using a single amino acid deletion technique, that lack of even one essential amino acid can have detrimental effects on the insect, especially in the larval stage (Wilkinson & Douglas, 2003). BCAA is very important, not only in protein synthesis but also in metabolic and regulatory roles. No studies have been done so far to investigate the role of branched-chain amino acids in B. tabaci, however, BCAA especially isoleucine has been reported to mediate glucose uptake and metabolism (Doi et al., 2003; Zhang et al., 2017). In B. tabaci, α-glucosidase genes (SUC1 & SUC2) are responsible for sucrose hydrolysis, a key sugar in phloem-sap that these insects solely depend on. It can be envisaged that downregulation of key α-glucosidase genes (SUC1 & SUC2), responsible for sucrose hydrolysis within the whitefly gut and BCAT will not only disrupt osmoregulation but also affect essential amino acid and energy production within cassava B. tabaci populations.

Conclusion

In conclusion, we have applied a system-biology approach to identify seven critical gene targets; three osmoregulation genes (AQP1, SUC1 and SUC2) and four symbiosis (argH (EC 4.3.2.1), lysA (EC 4.1.1.20), BCAT (EC 2.6.1.42) and dapB (EC 1.17.1.8)) that mediate sucrose hydrolysis, water recycling and terminal reactions of six essential amino acids in cassava whitefly (SSA1-SG1). These gene targets are highly expressed in the target organs where critical physiological processes such as osmoregulation and amino acid provisioning are regulated. These characteristics makes them good gene targets in the management of cassava whiteflies. More work is required to evaluate the impact of targeting these genes, singly or in combination, on the development and survival of B. tabaci. The best gene or gene combination causing the highest mortality of B. tabaci should be used in the development of stably transformed transgenic plants for use in integrated whitefly management programs in cassava growing areas. Management of cassava whiteflies will not only reduce the direct feeding damage caused by high whitefly population but will also reduce the spread of devastating viral diseases transmitted by these whiteflies and affecting food security in cassava growing areas.

Supplemental Information

Supplemental Information 1 SiRNA-Finder offtarget predictor results for the dsRNA sequences against AQP1 and SUC1 targets reported by Vyas et al., 2017.

Supplemental Information 2 Supplementary tables and figures.

Supplemental Information 3 RNASeq Analysis.

(A) Expression values for the transcripts with enriched expression in the SSA1-SG1 gut, (B) Interproscan annotation of transcripts with enriched expression in the SSA1-SG1 gut, and (C) Fischer’s exact enrichment test.

Supplemental Information 4 Single compartment metabolic model of portiera, primary endosymbiont of cassava whitefly.

Supplemental Information 5 Two compartment metabolic model of portiera, primary endosymbiont of cassava whitefly and its host, B tabaci (SSA1-SG1).

Supplemental Information 6 Calculation of the object function for the two compartment model of Portiera and B.tabaci (SSA1-SG1).

Supplemental Information 7 qPCR gene expression results.

Supplemental Information 8 MIQE checklist.

We thank Dr Jen Grenier (Cornell University) for advice in the design of the RNAseq analysis and Sharon Van Brunschot, Sophie Bouvaine, Nana Ankrah, Angela E Douglas, and Goncalo Silva for the support during the implementation of this study.

Additional Information and Declarations

Competing Interests

Author Contributions

DNA Deposition

Data Availability

The authors declare that they have no competing interests.

Tadeo Kaweesi conceived and designed the experiments, performed the experiments, analyzed the data, prepared figures and/or tables, authored or reviewed drafts of the article, and approved the final draft.

John Colvin conceived and designed the experiments, authored or reviewed drafts of the article, and approved the final draft.

Lahcen Campbell analyzed the data, authored or reviewed drafts of the article, and approved the final draft.

Paul Visendi analyzed the data, authored or reviewed drafts of the article, and approved the final draft.

Gareth Maslen analyzed the data, authored or reviewed drafts of the article, and approved the final draft.

Titus Alicai analyzed the data, authored or reviewed drafts of the article, and approved the final draft.

Susan Seal conceived and designed the experiments, analyzed the data, authored or reviewed drafts of the article, and approved the final draft.

The following information was supplied regarding the deposition of DNA sequences:

The raw sequencing reads for the RNA-Seq analysis are available in the European Nucleotide Archive repository: ERS4350849 for the whitefly gut sample, ERA2394091 for whitefly bacteriocytes, and ERS4350847, ERS4350848 for the whitefly whole body samples.

The following information was supplied regarding data availability:

The raw data is available in the Supplemental Files and at the European Nucleotide Archive.

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
