# Peer review of "In silico prediction of candidate gene targets for the management of African cassava whitefly (Bemisia tabaci, SSA1-SG1), a key vector of viruses causing cassava brown streak disease"

_PeerJ, doi:10.7717/peerj.16949_

## Round 0.1 · original submission · Major Revisions

One of the reviewers has recommended against publication of the manuscript but provided guidance on how it can be revised. The authors should carefully check the English grammatical mistakes throughout the manuscript.

**Language Note:** The Academic Editor has identified that the English language must be improved. PeerJ can provide language editing services - please contact us at [email protected] for pricing (be sure to provide your manuscript number and title). Alternatively, you should make your own arrangements to improve the language quality and provide details in your response letter. – PeerJ Staff

Reviewer 1 ·

Basic reporting

The manuscript is well described and executed. English language is satisfactory but need some improvements as indicated. Introduction needs some minor corrections & background to show context. All Figures are relevant, high quality, well labelled & described.
Here are few points to address;
L 24= In abstract write ‘Whiteuflies have a wide plant range and are globally important agricultural pests’. as ‘Whiteflies have a wide host plant range and are globally important agricultural pests.’
L 26- 28= rewrite as “Efforts are ongoing to identify target genes to develop novel management options against the whiteflies for the effective production of cassava, an important crop for food security more than 800 million people in Sub-Saharan Africa.”
L45= B. tabaci Full genus name at first mention, preferably along with whiteflies at the start of the abstract
Abstracts need to be as 1 para
L54= Bemisia. tabaci, should be written as Bemisia tabaci (no full stop after genus name need to check in all MS)
L104= In silico should be in silico
L296= B. tabaci should be italics. Check in all MS
L315= whiteflies/ B. tabaci, keep uniformity
L406-408= rewrite
Discussion need improvements, the authors should avoid introduction or literature review style.
References need to be checked for uniformity as L591= Bemisia tabaci should be italics
Full genus names need to be given in all tables and figures
Authors included important tables and figures in the main manuscripts and some as supplementary material.

Experimental design

The manuscript falls within the scope of the journal. Rigorous investigation performed to a high technical & ethical standard. Methods described with sufficient detail & information to replicate.

Validity of the findings

All underlying data have been provided; they are robust, statistically sound, & controlled.
Conclusions are well stated, linked to original research question.

Additional comments

NA

Reviewer 2 ·

Basic reporting

This research paper would greatly benefit from a restructured introduction that highlights the existing research gaps, clearer delineation of research objectives and novelty, proofreading and editing of the methodology section, concise presentation of results, and a more focused and well-cited discussion section. These improvements will enhance the paper's overall quality and make it more informative and accessible to the readers.

Experimental design

The introduction of the research paper lacks appropriate support from the literature, which diminishes its effectiveness in providing context for the study. It is advisable to rearrange the introduction as follows:
The first paragraph should highlight the existing gaps in the research related to the management of African cassava whitefly and its role as a vector of Cassava Brown Streak Disease. The subsequent paragraph should introduce the main topic of the study, providing a concise overview of the research objectives and the novelty of the research. The next paragraph should emphasize the significance of the research, its potential implications, and the unique contributions it brings to the field. It is essential to state the research objectives and novelty more clearly, providing a well-defined roadmap for the reader.

Validity of the findings

The methodology section contains typographical errors that need correction. These errors detract from the overall quality of the paper. A thorough proofreading and editing of this section is necessary to maintain the paper's credibility.
The results section appears to be excessively lengthy, which may overwhelm the reader. It would be beneficial to present the key findings concisely, focusing on the most significant results while providing the option for readers to delve deeper into the supplementary material for additional details.
The discussion section should emphasize the findings of this study and connect them with appropriate citations from previous literature. This connection with existing research is vital for contextualizing the study and demonstrating its relevance in the field. The current discussion may benefit from a more structured approach, highlighting the key findings and their implications in a logical order. Additionally, integrating relevant citations from previous literature will strengthen the discussion and provide a more comprehensive perspective on the research.

---

## Round 0.2 · Major Revisions

Authors are advised to revise manuscript very carefully, as per the suggestions.

Reviewer 1 ·

Basic reporting

Authors need to revisit my previous comments and apply on the whole manuscript, not only the mistakes highlighted.
There should be no full stop at the end of the title.
Line 152 and so many other places= wholebody should be whole body?
SingleGeneDeletion of Single Gene Deletion?
Haemolymph or hemolymph. Use US English
Discussion need to be restructured. Start with your results without any introduction
References need to be rechecked to match journal style. For instance, some Journal names appeared as abbreviations and some as full?
Line 806= Bemisia tabaci still not italicized although pointed out in my previous comments and authors have claimed that they corrected in all. Same issue in Fig 1. So I suggest to revisit my previous comments.

Experimental design

Fig 7, Y-error bars are seen. See some appropriate statistical analysis.

Validity of the findings

Conclusion need to be written in a scientific way. It should not look as an introduction

Reviewer 2 ·

Basic reporting

Improved from previous version of manuscript.

Experimental design

Methodology is improved from previous version, can be traced easily.

Validity of the findings

Results and discussion are written well and improved from previous version of the manuscript.

Additional comments

The manuscript has improved from the previous version but still few formatting errors were found that need to be corrected.
1. Use SI units only, in line 155, write µL.
2. Check for typographic errors, add space after comma in line 289 & 290.
3. Use percentage or Unit with last digital value only while writing 2 or more values in row. As in line 292.
4. Always give space between value and unit. As in line 150. Also correct unit of temperature (oC) in line 150, 267, 277 and 278.
5. Check whole manuscript for such formatting and typographical errors.
6. References style is not same, either use abbreviated form or full form for journal name. Use same reference style as per journal’s guidelines.
7. Italicize scientific names even in the references list.

---

## Round 0.3 · accepted · Accept

Authors have revised the manuscript as per reviewers suggestions. Therefore, I recommend that manuscript can be accepted for publications.

The Section Editor noted:

> In the proof, I am not sure if "compasation" is a word, and maybe the authors mean "compensation"? (only found once in a Figure legend)

Reviewer 1 ·

Basic reporting

The authors have improved the MS.

Experimental design

OK

Validity of the findings

OK

Reviewer 2 ·

Basic reporting

It is fine

Experimental design

It is fine

Validity of the findings

It is fine.

Additional comments

Authors have made changes as per instructions. Manuscript can be considered for publication.